# Effects of Exponential N Application on Soil Exchangeable Base Cations and the Growth and Nutrient Contents of Clonal Chinese Fir Seedlings

**DOI:** 10.3390/plants12040851

**Published:** 2023-02-14

**Authors:** Renjie Wang, Yong Wang, Zeyao Zhang, Huibiao Pan, Liufeng Lan, Ronglin Huang, Xiaojun Deng, Yuanying Peng

**Affiliations:** 1Guangxi Forestry Research Institute, Nanning 530002, China; 2School of Automation, Central South University of Forestry and Technology, Changsha 410004, China; 3Huangmian State-Owned Forest Farm in Guangxi, Liuzhou 545000, China; 4College of Arts and Sciences, Lewis University, Romeoville, IL 60446, USA

**Keywords:** N addition, base cations, plant growth, soil acidification, Chinese fir seedlings

## Abstract

Nitrogen (N) is an essential macronutrient for plant function and growth and a key component of amino acids, which form the building blocks of plant proteins and enzymes. However, misuse and overuse of N can have many negative impacts on the ecosystem, such as reducing soil exchangeable base cations (BCs) and causing soil acidification. In this research, we evaluated clonal Chinese fir (*Cunninghamia lanceolata* (Lamb.) Hook) seedlings grown with exponentially increasing N fertilization (0, 0.5, 1, 2 g N seedling^−1^) for a 100-day trial in a greenhouse. The growth of seedlings, their nutrient contents, and soil exchangeable cations were measured. We found that N addition significantly increased plant growth and N content but decreased phosphorous (P) and potassium (K) contents in plant seedlings. The high nitrogen (2 g N seedling^−1^) treated seedlings showed a negative effect on growth, indicating that excessive nitrogen application caused damage to the seedlings. Soil pH, soil exchangeable base cations (BCs), soil total exchangeable bases (TEB), soil cation exchange capacity (CEC), and soil base saturation (BS) significantly decreased following N application. Our results implied that exponential fertilization resulted in soil acidification and degradation of soil capacity for supplying nutrient cations to the soil solution for plant uptake. In addition, the analysis of plants and BCs revealed that Na^+^ is an important base cation for BCs and for plant growth in nitrogen-induced acidified soils. Our results provide scientific insights for nitrogen application in seedling cultivation in soils and for further studies on the relationship between BCs and plant growth to result in high-quality seedlings while minimizing fertilizer input and mitigating potential soil pollution.

## 1. Introduction

Nitrogen (N) plays a key role in plant growth and N fertilizer is used to enhance the growth and productivity of crops or trees [1,2,3] but is also a limiting nutrient for plants [4]. Nutrient loading techniques based on exponential fertilization have shown that exponential fertilization was more consistent with the N required for plant growth patterns than conventional fertilization [5,6,7]. Compared to constant-rate nutrient loading of conventional fertilization, the rate of nutrient supply increases exponentially and exceeds the rate of seedling growth, minimizing the risk of nutrient toxicity in seedlings [8,9]. It is important to note that N additions may negatively impact ecosystem functions [10,11].

A common indicator for determining the availability of most plant nutrients is the measurement of the cation exchange capacity (CEC) in the soil [12]. The cation exchange capacity, or exchange rate, describes the number of cations in the soil solution that are exchangeable, and therefore available for the uptake by plant roots. Plants extract the positively charged cations from the soil solution by exchanging them for positively charged hydrogen (H^+^). Soil exchangeable base cations (BCs), mainly including calcium (Ca^2+^), magnesium (Mg^2+^), potassium (K^+^), and sodium (Na^+^), which are essential cations for plant growth and soil ecosystem stability [12,13]. Cations, such as Mg^2+^, are essential for photosynthesis and energy storage, and they are involved in the regulation of plant physiology and biochemistry [14,15]. Soil BCs play an important role in the regulation of soil acidification. This is because the exchangeable BCs can not only be absorbed by soil colloids but can also provide exchange sites for H^+^ and alleviate ecosystem damage [13,16]; this is essential for maintaining soil nutrients and buffering soil acidification [13,17]. However, accelerating industrialization and use of N fertilizer has led to increased atmospheric N deposition [18,19], thereby accelerating soil acidification processes [1,10], reducing the capacity of soil exchangeable BCs [13], decreasing soil buffering capacity [20], and lowering plant productivity [21,22].

Chinese fir (Cunninghamia lanceolata (Lamb.) Hook) is an evergreen conifer species, which has been widely cultivated as an important timber species for over 1000 years in the sub-tropical region of China. The Chinese fir forests account for 60%–80% of the total area of timber plantations in southeast China and their annual timber production accounts for 20%–25% of the national commercial timber output [23]. This tree species is considered an economic timber species because of its fast growth, good quality material, strong wood property, hardiness, versatile use, and high timber yield per unit, which is used extensively for pulp, paper, and lumber [5,24,25]. Human activities have led to major increases in global emissions of nitrogen (N) to the atmosphere [26]. Several studies have shown that N deposition or/and fertilization have increased significantly in recent years [27,28,29]. The main concern is the damaging effect of excessive nitrogen loads in many contrasting ecosystems and toxicity to plant growth [30]. Many previous studies have looked at the effects of N fertilization on plant growth [2,5,9] or exchangeable BCs [12,13,20] in soils. However, the effects of N application on soil BCs and plant growth, and the relationship between soil BCs and plant growth are still unclear. In addition, Chinese fir tree seedlings are commonly grown in containers before planting in the fields. During this period, gardeners or planters usually add N to promote seedling growth, but few studies have focused on the effects of N on plant growth and BCs [5].

Unlike field experiments, the effects of environmental heterogeneity on experiments are often reflected at a very small scale, greenhouse-controlled experiments specifically avoid this problem [31]. In this study, Clonal Chinese fir seedlings were treated with four concentrations of N exponential fertilization growing in a greenhouse. The nutritional contents of seedlings and soil exchangeable cation (BCs) were measured. We hypothesized that (1) N addition would significantly increase the Chinese fir seedlings growth at the low N concentration levels and decrease growth at high N concentration application, (2) N addition would increase the N content but decrease phosphorous (P) and potassium (K) contents in plant seedlings, and (3) the N application would decrease soil pH, soil exchangeable base cations (BCs), soil total ex-changeable bases (TEB), soil cation exchange capacity (CEC), and soil base saturation (BS). The objectives of the study were (i) to explore the effects of treatments of N exponential fertilization on the growth and nutrient contents of Chinese fir seedlings, (ii) to reveal the effects of N exponential fertilization on soil chemical properties including exchangeable BCs, soil total exchangeable bases (TEB), soil cation exchange capacity (CEC), and soil base saturation (BS) together with the relationships among them, and (iii) to analysis the relationship between soil exchangeable BCs and seedling growth.

## 2. Materials and Methods

### 2.1. Study Areas

The pot experiments were conducted in a cultivated greenhouse located in Guangxi Forestry Research Institute, Nanning city, Guangxi Zhuang Autonomous Region (108°21′ E, 22°55′ N). Circular pots with inner diameter of 25 cm and a height of 35 cm were filled with clay loam red soil (about 27 kg, 1/3 water content) derived from local Chinese fir (*Cunninghamia lanceolata* (Lamb.) Hook) forest. One-year-old clonal Chinese fir seedlings from the same nursery site with similar height and base diameter were used for the experiments. The Chinese fir seedlings were transplanted into pots and one seedling kept in each pot and the pots were randomized each week to prevent the effects of light.

### 2.2. Experimental Design

A complete randomized design (CRD) with four levels of exponential N application treatment was performed in the experiment. The four treatment levels of exponential N addition treatments were the control (without N addition, CK), a low level of N addition (0.5 g N sapling^−1^, L1), a medium level of N addition (1 g N sapling^−1^, L2), and a high level of N addition (2 g N sapling^−1^, L3) with 10 replicates (10 pots for each treatment level), so that a total of 40 pots (4 × 10) was used in the experiment. The first fertilization was carried out on 15 May 2021, after the seedlings in the pots had grown steadily (no seedling died within three days). N fertilization was then applied every 10 days for a total of 10 applications (last fertilization on 22 August). Sample saplings were collected and measured ten days after the end of the exponential fertilization (1 September). Exponential fertilization was based on an exponential model to determine the amount of nitrogen per application [5], which was calculated by the following formula:*N_T_* = *N_S_* (*e^rt^* − 1)(1)
*N_t_* = *N_S_* (*e^rt^* − 1) − *N*_*t*−1_(2)
where *r* is the relative rate of addition required to increase *Ns* (the initial level of the nutrient), to the final level *N_T_* + *Ns*, and *N_t−_*_1_ is the cumulative amount of fertilizer added including the last application. The amount *Ns* (72.19 mg N) was determined by chemical analysis of saplings 3 days before the start of fertilization. Knowing *N_T_* (0.5, 1, or 2 g N) and *N_S_*, *r* by Equation (1) to determine the number of fertilizations (*t* = 10), we can then calculate the exact amount of each fertilization by Equation (2). N was added in the form of urea (CH_4_N_2_O) solution, and the details of the fertilization process are shown in Figure 1.

### 2.3. Measurements

The seedling height from ground to apical meristem was measured using linear tape and the base diameter was measured in two vertical directions with a vernier caliper; the mean was calculated and used for analysis [32]. The measurements were taken on the first day and the last day of the experiment. In addition, the weekly measurements were taking between 8 am to 10 am during the experiment. The increments in height and base diameter of seedlings were calculated from the differences of two measurements (the first day and the last day of the experiment). At the end of the experiment, all seedlings were harvested to measure biomass and nutrient content. Each individual seedling was divided into root, stem and leaves for biomass measurements and chemical analysis in the laboratory. Plant samples were oven dried at 105 °C for 15 min and then at 75 °C for 48 h until constant weight for determining the biomass using an analytical balance. All leaves were ground and passed through a 40-mesh sieve for nitrogen (N), phosphorus (P), and potassium (K) analysis in the laboratory. Total N and P were determined according to the standard methods and the total potassium (K) was analysis by plasma emission spectrophotometry after dry ashing and extraction with HCl–HNO_3_ [33].

Three soil samples in each pot were collected after harvesting of the plant seedlings. Each soil sample was passed through a 2-mm sieve to remove roots and stones, and then air-dried. Soil pH was measured with a glass electrode at a water: soil ratio of 1:2.5 (*w*/*v*), and soil organic carbon (SOC) was measured by the potassium dichromate oxidation heating method. Total nitrogen (TN) was determined by spectrophotometry after digestion with K_2_Cr_2_O_7_–H_2_SO_4_ using an automatic Kieldahl instrument (K9860, Hanong, Kaiyuan, China). The NH_4_^+^-N and NO_3_^–^-N were measured by extraction using an automatic continuous flow analyzer (Autoanalyzer 3, Seal analytical, Norderstedt, Germany). Determination of total phosphorus (TP) was carried out by the molybdenum blue colorimetric method, after hydrolysis and diffusion of NaOH, using a UV-Vis spectrophotometer (UV-6800A, ZhuoGuang, Jinan, China). Total potassium (TK) was determined by spectrophotometry after wet digestion with HF–HClO_4_ by an atomic absorption spectrophotometer (AA-7000, Shimadzu, Kyoto, Japan). The exchangeable acid (EA) was measured by potassium chloride exchange—neutralization titration. The exchangeable BCs (K^+^, Ca^2+^, Mg^2+^ and Na^+^) were measured by atomic absorption spectrophotometry using the NH_4_Cl–C_2_H_5_OH method (AA-7000, Shimadzu, Kyoto, Japan) [34].

### 2.4. Statistical Analyses

One-way analysis of variance (ANOVA) was used to assess the effects of different levels of exponential N application on increments of height, base diameter, and nutrient element content of each organ (root, stem, leaf) of seedlings and soil physicochemical properties. Tukey’s HSD was used to test significant differences among different N addition treatments (*p* < 0.05). The Shapiro–Wilk test was used for the data analysis, and the original data were log-transformed to satisfy the normality and homoscedasticity assumptions of ANOVA. Pearson’s correlation test was used to analyze the relationship among different parameters and the results were displayed by R package ‘ggcorrplot’ (version: 0.1.3). Considering collinearity, soil TEB, CEC, and BS were not included in the correlation analysis. The means of N application effects were compared by a Tukey–Kramer test using the SAS program. Multiple comparisons were conducted to test the differences among treatments in response to plants growth and nutrient content of plant seedlings including N, P, and K (*p* < 0.05). To examine the relationship between BCs and plant growth, a stepwise regression model (SRM) was used to select factors affecting plant growth with the best-fit model chosen using Akaike’s Information Criterion (AIC). Statistical analyses were conducted using the SAS statistical package [35], and the statistical programming language R [36].

## 3. Results

### 3.1. Effects of N Addition on Seedling Growth and Nutrient Content of Plants

Compared with CK (no N fertilization), N addition was beneficial to plant growth (Table 1). The increments of ground diameter and height increased with increasing N fertilization concentrations, except in L3. Biomass of root, stem, and leaf and the total biomass of seedlings showed an increasing tendency with N addition. The biomass of each component of the plant seedlings had significant effects in response to the N application treatments (*p* < 0.05) except for root biomass. In general, seedlings grew best in L2 treatments. A similar growth state of the seedlings was observed between CK and L3 treatments, indicating the negative effect of high N addition.

Nutrient elements of seedlings were significantly different between CK and N addition treatments (*p* < 0.05; Figure 2). With increasing N fertilization concentrations, the N content in the organs of the seedling increased, especially in the roots and leaves. The N content in roots at L3 reached 43.21 ± 0.15 mg·g^−1^, while the N in stems decreased to 21.50 ± 0.14 mg·g^−1^ (Figure 2a). Conversely, P and K contents decreased after N addition, except for K in the roots. The content of P was the highest in roots at CK (3.30 ± 0.06 mg·g^−1^), and the lowest was in leaves at L3 (1.65 ± 0.02 mg·g^−1^) (Figure 2b). Both the minimum and maximum K contents occurred at CK, which were 7.95 ± 0.09 mg·g^−1^ in roots and 13.85 ± 0.09 mg·g^−1^ in leaves, respectively (Figure 2c).

### 3.2. Effect of N Addition on Soil Physicochemical Properties

Soil physicochemical properties showed a significant difference between N addition and CK treatments (*p* < 0.05) (Figure 3). Soil pH, BCs, TEB, CEC, and BS decreased, but SOC, TN, NO_3_^−^, NH_4_^+^, and EA increased with increasing N fertilization concentrations. No significant differences of EA and CEC were observed among all treatments, with the corresponding maximum values of 145.73 ± 0.22 cmol·kg^−1^ and 157.00 ± 0.67 cmol·kg^−1^, and minimum values of 142.26 ± 0.85 cmol·kg^−1^ and 154.40 ± 0.27 cmol·kg^−1^, respectively. BCs were greatly reduced with increasing N fertilization (except for Ca^2+^ and Na^+^ in L1), which led to an increase of EA and a stable status of CEC. Compared to CK, N fertilization resulted in decrease of TP and increase of K, but there were no significant differences between these two elements among N addition treatments (Figure 3).

### 3.3. Relationships among Different Soil Parameters

The four types of BCs (Ca^2+^, Mg^2+^, K^+^, and Na^+^), pH, and TP showed a significant positive correlation with each other, while BCs showed a significant negative correlation with SOC, TN, NO_3_^−^, and NH_4_^+^ (Figure 4). pH was negatively correlated with SOC, TN, NO_3_^−^, NH_4_^+^, and EA, which were directly influenced by N addition. NO_3_^−^ had a closer relation (high coefficient) with other soil parameters than NH_4_^+^. Soil K was not significantly correlated with any other soil parameters except soil pH, SOC, and TN.

### 3.4. The Correlations between Plant Growth and BCs

In the SRMs of diameter, K^+^, Na^+^, and Mg^2+^ were the fixed variables in the optimization model (Table 2), and only Na^+^ was significantly positively correlated with diameter (*p* = 0.04). K^+^, Na^+^, and Mg^2+^ were also selected as the fixed variables in the optimization SRM of height, and K^+^ was a significant factor negatively affecting plant height (*p* = 0.001), while Na^+^ was significantly positively correlated with plant height (*p* = 0.01). There were two fixed variables (K^+^ and Na^+^) in the optimization SRM of biomass, which were significantly negatively and positively correlated with biomass, respectively (Table 2). Among the exchangeable BCs, the cations of K^+^, Na^+^, and Mg^2+^ may be the important BCs affecting the growth of seedlings under N fertilization, especially Na^+^.

## 4. Discussion

### 4.1. Growth and Nutrient Contents of Seedlings

Our results showed that exponential N addition enhanced the growth of Chinese fir saplings. The diameter, height, and biomass of the saplings were significantly higher in N adding treatment than in control but the enhanced effect was weaker in high N treatment (L3). The high nutrient availability reduced the root biomass as well in the research. This is because roots confine their growth with sufficient availability of nutrients as they do not have to penetrate deeply in search of the nutrients [37,38,39]. Previous studies have demonstrated that exponential fertilization improved plant growth by nutrient loading [2,5,8,13]. N fertilization increased N content, but not P and K contents in leaves. It was clear that the N addition led to an increase in N content in plants. In this study, P generally showed a decreasing trend with N addition, which was likely attributable to the following two points. One was that the N addition led to the imbalance of the N:P ratio in soil, thus affecting the uptake of P by plants [31,40]. The other was that the addition of N caused a dilution effect on P, leading to a decrease of its content in plants. The K content increased with N addition in root but decreased in stem and leaf, suggesting absorption and accumulation of K element in the root. However, not all the studies showed plant growth was positively associated with N addition. Experiments of western hemlock [9] and Douglas-fir [6] showed that exponentially fertilized seedlings did not differ significantly in growth and nutrient contents. We also found that the growth of seedlings was influenced in the high N addition treatments (L3). This was because higher nutrient levels in seedlings may increase their susceptibility to moisture stress, frost damage, and herbivory [9,41], while N addition-induced soil acidification also influences seedling growth.

### 4.2. Response of Soil Parameters to N Addition

In this study, there were significant differences in soil physicochemical properties between the N addition treatment and the control treatment CK (*p* < 0.05). Soil pH, BCs, TEB, CEC, and BS decreased with increasing N fertilization, but SOC, TN, NO_3_^−^, NH_4_^+^, and EA increased under N addition treatments. Soil TP reduced and K increased in the N addition treatments compared to CK. Many studies have illustrated that N fertilization can result in decreasing soil pH [3,13,17,20,22]. Previous studies showed that N addition could lead to increasing hydrolysis of BCs and release of H^+^ in the soil solution [42,43]. H^+^ had more proton competitiveness to BCs for cation exchange sites which resulted in lower pH and soil BC depletion [13,44]. Decreasing BC content could cause lower TEB, CEC, and BS, and a low BS (<10%) also indicated an acidified context [13,45]. These processes might be the reason why the pH showed significant positive correlation with four BCs. On the other hand, N fertilization increased the nitrification process, which caused a decrease in pH and an increase in EA [46]. Excess of N input led to an increase of NO_3_^−^ leaching from bulk soils into soil solution [10,47,48]. However, leaching loss of NO_3_-N was often accompanied by leaching of Ca^2+^, Mg^2+^, and other cations [47]. In addition, studies demonstrated that increase of NH_4_^+^ would give a stronger bond strength with soil and could exchange the BCs adsorbed by the soil (primarily for K^+^ and Na^+^) and increase the loss of soil BCs [49]. Our results showed both NO_3_^−^ and NH_4_^+^ increased by N addition, which further caused the decrease of soil BCs. These phenomena of small change of CEC under different N concentrations were ascribed to (1) the magnitude of change in BCs being much smaller than that in CEC and (2) the cations of Al^3+^ and Fe^2+^ making major contribution to soil CEC [17], which were restricted in the pot experiments.

For other nutrients, SOC increased after N fertilization, which promoted plant productivity. Stimulation of microbial activity in soil by N addition may play a key role in this process [50,51]. There was a coupling relationship between N and P resulting in N:P imbalance due to N deposition [31,40]. Similar results were also observed in other studies [13,46]. We noticed inconsistent changes of K content in soil and plants as well in our research. Although the decrease in K^+^ resulted in decrease in the K content in stems and leaves, the K content increased in roots after N addition which was consistent with the changes in soil K. This may be related to the complex conversion between K and K^+^ [52] while our research might explain the interrelationships among the soil parameters.

### 4.3. Effects of BCs on Chinese Fir Seedling Growth

Three SRMs of plant growth, K^+^, Na^+^, and Mg^2+^ were set as fixed variables in the diameter and height optimization model. Only K^+^, and Na^+^ were in the biomass optimization model. Na^+^ was significantly positively correlated with both diameter (*p* = 0.04) and height (*p* = 0.01) indicating that Na^+^ may be a key cation for plant growth in acidified soil. Na^+^ is not a nutrient directly involved in plant metabolism but plays a key role in soil acidity buffering along with the other BCs [13,53]. Soil acidification can lead to an imbalance of soil elements that ultimately affects plant growth [1,20]. Therefore, Na^+^ may be more closely related to plant growth in the pot experiment in acidic soil (lower pH and BS) induced by nitrogen addition. Both N and K^+^ can increase plant productivity, but N addition resulted in the decrease in K^+^ in this study. In addition, K^+^ was reduced more than Na^+^ in our study because K^+^ was easily taken up by the plant roots rather than remaining in the soil [52]. Therefore, our results suggest a negative correlation between K^+^ and plant growth. Further research should focus more on how BCs directly affect plant growth.

## 5. Conclusions

In summary, our study demonstrated the exponential application of N fertilizer significantly affected the Chinese fir seedlings growth and nutrient concentrations in the plant organs. When compared to the control, the low N addition (0.5 and 1 g N seedling^−1^) significantly increased the Chinese fir seedlings growth and the N content in plants but decreased phosphorous (P) and potassium (K) contents in plant seedlings. However, the high N addition (2 g N seedling^−1^) decreased the growth of Chinese fir seedings because the excessive N application damaged the metabolism of the seedlings. N application significantly decreased soil physicochemical properties, including soil pH, soil exchangeable base cations (BCs), soil total exchangeable bases (TEB), soil cation exchange capacity (CEC), and soil base saturation (BS). Our hypothesis was supported by the results. In addition, the N exponential fertilization can lead to acidification and degradation of the cation exchange capacity in the soil while Na^+^ was the most important base cation for soil BCs and for plant growth in N-induced acidified soils. Our results provide a solid scientific basis for improved understanding of N fertilizer application in plant seedling cultivation.

## Figures and Tables

**Figure 1 plants-12-00851-f001:**
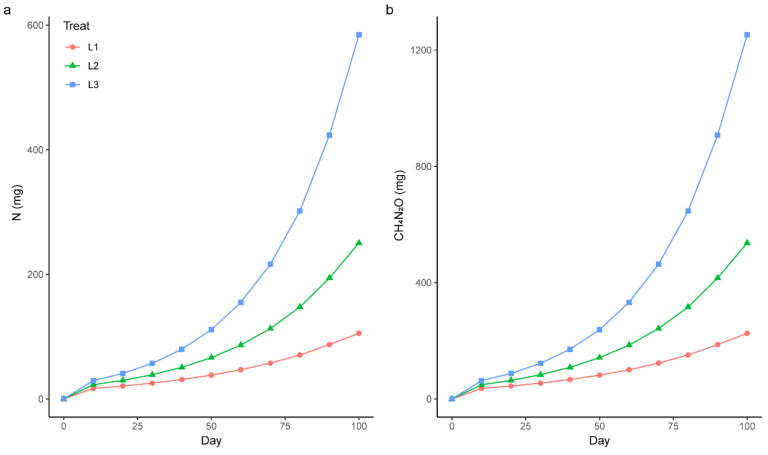
The schedule of N fertilization during the 100 days study; (**a**) is the cumulative usage of N, and (**b**) is the cumulative usage of CH_4_N_2_O (urea). Different colors and symbols indicate the different treatments.

**Figure 2 plants-12-00851-f002:**
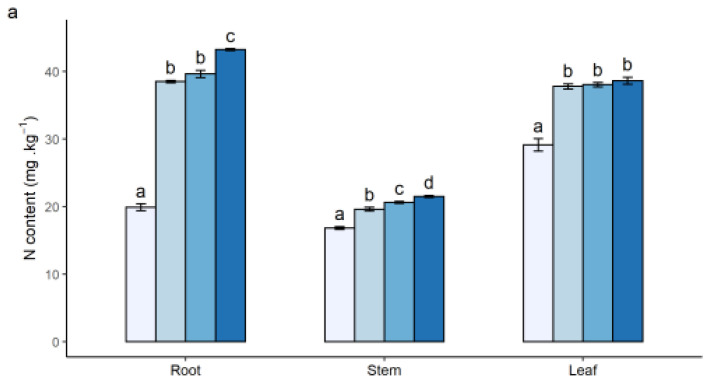
The content of nutrient elements in three organs (roots, stems, and leaves) of plants in response to different N addition treatments. N content (**a**), P content (**b**), and K content (**c**). Bars represent mean value with standard error (±SE). Different colors indicate the different treatments. Different letters indicate the significance among different levels of N application (*p* < 0.05).

**Figure 3 plants-12-00851-f003:**
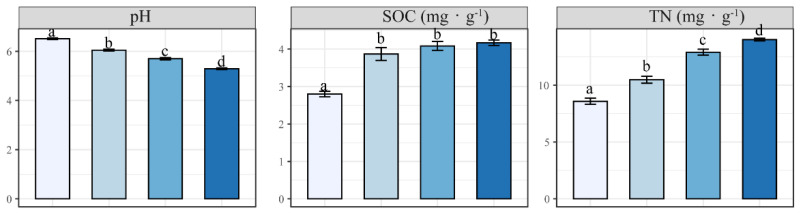
The concentration of soil parameters in different N addition treatments. Bars represent mean value with the standard error (±SE). Different colors indicate the different treatments. Different letters indicate the significance among different levels of N application (*p* < 0.05).

**Figure 4 plants-12-00851-f004:**
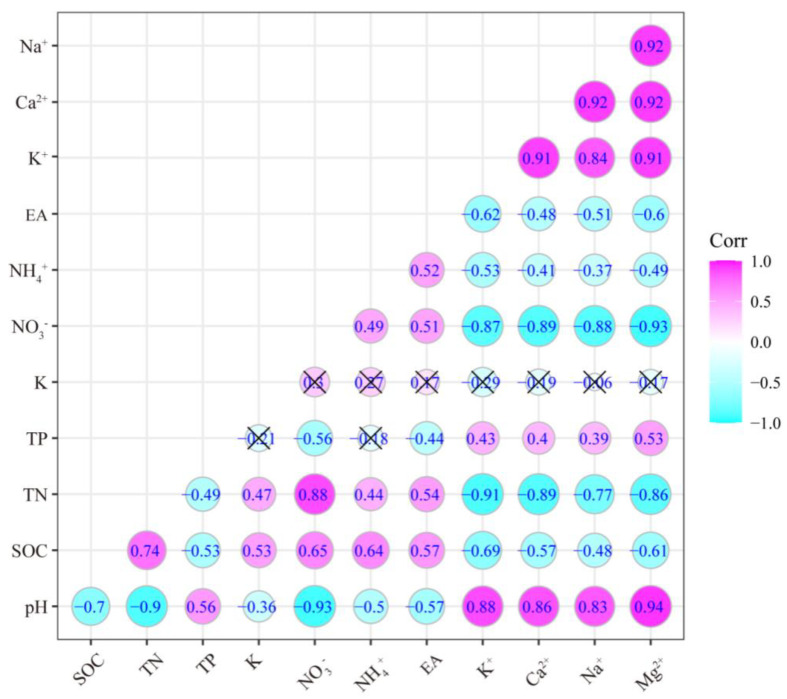
The correlation among soil parameters by Pearson’s correlation test. “Cyan” indicates negative correlations and “magenta” indicates positive correlations. The circle size is proportional to the Pearson’s coefficient. A cross means statistical insignificance (*p* > 0.05).

**Table 1 plants-12-00851-t001:** Growth of seedlings in different N fertilization treatments. Diameter and height are increments compared to the initial size. Data are given the mean ± SE (*n* = 10).

Treat	Diameter (cm)	Height (cm)	Root (g)	Stem (g)	Leaf (g)	Total Biomass (g)
CK	4.13 ± 0.12a	15.90 ± 0.34a	5.69 ± 0.06a	5.31 ± 0.05a	5.37 ± 0.06a	16.37 ± 0.13a
L1	4.64 ± 0.10b	19.47 ± 0.29b	7.34 ± 0.10b	5.86 ± 0.06b	6.09 ± 0.05b	19.29 ± 0.15b
L2	5.62 ± 0.06c	23.26 ± 0.38c	7.67 ± 0.09b	6.28 ± 0.08c	6.20 ± 0.11b	20.14 ± 0.15c
L3	4.88 ± 0.13b	20.59 ± 0.37b	5.89 ± 0.13a	5.90 ± 0.07b	5.76 ± 0.07c	17.54 ± 0.21d

Note: Different letters indicate significant differences among different levels of N application (*p* < 0.05).

**Table 2 plants-12-00851-t002:** The summary of SRMs for plant growth (diameter, height, and biomass) and BCs.

Variate	Fixed	Estimate	SE	*t*-Value	*p*-Value	AIC
Diameter	Intercept	8.00	1.05	7.60	<0.001	−50.45
	K+	−2.95	1.86	−1.58	0.12	
	**Na**+	**9.23**	**4.33**	**2.13**	**0.04**	
	Mg2+	−0.46	0.26	−1.75	0.09	
Height	Intercept	33.43	3.97	8.42	<0.001	55.93
	**K**+	**−24.61**	**7.07**	**−3.48**	**0.001**	
	**Na**+	**44.55**	**16.03**	**2.78**	**0.01**	
	Mg2+	−1.39	0.98	−1.42	0.16	
Biomass	Intercept	20.90	0.87	24.12	<0.001	28.02
	**K**+	**−14.92**	**3.91**	**−3.82**	**<0.001**	
	**Na**+	**26.89**	**8.30**	**3.24**	**0.003**	

Note: The optimal model was screened by the AIC. Bold indicates statistical significance (*p* < 0.05).

## Data Availability

The data presented in this article are available on request from the corresponding authors.

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
