# Peer review of "Effects of Exponential N Application on Soil Exchangeable Base Cations and the Growth and Nutrient Contents of Clonal Chinese Fir Seedlings"

_plants, 2023, doi:10.3390/plants12040851_

Round 1

Reviewer 1 Report

This study examines the effect of incremental N fertilization on soil exchangeable base cations and the growth and nutrient contents of clonal Chinese fir seeding in a greenhouse setting. Given the importance of the Chinee fir, both economically and ecologically, in the context of China, and the lack of research in this study field, this study has merits for publication in Plants. It is well written and presented with sound experimental designs and appropriate analyses. That said, the following minor concerns need to be addressed before its acceptance for publication. 

1. Line 124, The measurements were taken every day during the experiment from the first day until the last of the experiment. Please specify at what time every day those measurements were taken.

2. Line 154, Tukey’s HSD was used to test significant differences among different N addition treatments (p < 0.05). Please justify the use of Tukey's HSD.

3. Please add a conclusion section, following the discussion section.

4. Please address research limitations, if any, in the conclusion section

5. This manuscript needs a thorough editing and proofreading. For example, the total potassium (K) was analysis by (line 133)-->was analyzed; leaching loss of NO3-N were often (line 267)-->was often; which were mainly contribute to CEC [17] (line 274)-->which mainly contributed to. Please check throughout the text. 

Author Response

Responses to Reviewer #1

This study examines the effect of incremental N fertilization on soil exchangeable base cations and the growth and nutrient contents of clonal Chinese fir seeding in a greenhouse setting. Given the importance of the Chinee fir, both economically and ecologically, in the context of China, and the lack of research in this study field, this study has merits for publication in Plants. It is well written and presented with sound experimental designs and appropriate analyses. That said, the following minor concerns need to be addressed before its acceptance for publication. 

Re: Thank you so much for the reviewer’s valuable review comments and excellent suggestions on our manuscript (MS), we greatly appreciate it. We have revised it, please see the revision. 

  1. Line 124, The measurements were taken every day during the experiment from the first day until the last of the experiment. Please specify at what time every day those measurements were taken.

Re: We have revised it in the revision The measurements were taken on the first day and the last day of the experiment. In addition, the weekly measurements were taken between 8am to 12 pm during the experiment. Please see the revision. 2. Line 154, Tukey’s HSD was used to test significant differences among different N addition treatments (p < 0.05). Please justify the use of Tukey's HSD.

Re: We have added the information regarding the Tukey-Kramer test. “The means of N application effects were compared by a Tukey-Kramer test using SAS program. Multiple comparisons were conducted to test the differences among treatments in response to plants growth and nutrient content of plant seedlings including N, P and K (p < 0.05)”. Please see the revision.

  1. Please add a conclusion section, following the discussion section.

Re: We have added a conclusion paragraph at the end of the “Discussion” section in the revision.

  1. Please address research limitations, if any, in the conclusion section

Re: We have indicated that “Our results provided a solid scientific basis for further understanding of N fertilizer application in plant seedling cultivation” at the end of conclusion in the revision.

  1. This manuscript needs a thorough editing and proofreading. For example, the total potassium (K) was analysis by (line 133)-->was analyzed; leaching loss of NO3-N were often (line 267)-->was often; which were mainly contribute to CEC [17] (line 274)-->which mainly contributed to. Please check throughout the text

Re: We have double checked all the above information and have revised corresponding sentences in the revision.

Reviewer 2 Report

-The paper needs the hypothesis before the objectives.

- English needs minor improvement.

- Concrete Conclusions should be included after discussion.

- Revise tge paper as per suggestions and comments made in the manuscript.

Author Response

Comments and Suggestions for Authors

-The paper needs the hypothesis before the objectives.

Re: Good suggestion. We have inserted our hypothesis for this study before the objectives in the ‘Introduction’ section in the revision.

- English needs minor improvement.

Re: We have double checked and revised the entire MS carefully, and the MS has been improved lots.

- Concrete Conclusions should be included after discussion.

Re: Yes, the conclusion paragraph has been added at the end of the ‘Discussion’ section. 

- Revise the paper as per suggestions and comments made in the manuscript.

Re: We already have double checked the entire MS and revised the MS according to the two reviewer’s comments and suggestions.

Reviewer 3 Report

Well-written manuscript and well-presented research.

My greatest concern is about the lack of conclusions or brief summary at the end of the paper - consider adding a paragraph that would summarise the work you had done

L126 which two measurements exactly you took to calculate the increment: consequtive ones or the first and the last day?

L139 WITH missing before spectrophotometry

L151 support the choice of ANOVA with results of Shapiro-Wilk test, if the distribution diverses from the normal one, recalculate using Kruskal-Wallis test

L157 would be good to have correlation coefficient values here so that this colinearity would be obvious for the reader

figures 2, 3 - bars represent mean, but SE is shown by WHISKERS - add this please

Author Response

Responses to Reviewer #2

My greatest concern is about the lack of conclusions or brief summary at the end of the paper - consider adding a paragraph that would summarise the work you had done

Re: We greatly appreciate the reviewer’s valuable review comments and suggestions on the manuscript (MS). We have added the conclusions paragraph at the end of discussions in the revision.

L126 which two measurements exactly you took to calculate the increment: consequtive ones or the first and the last day?

Re: The measurements were taken on the first day and the last day of the experiment. . In addition, the weekly measurements were taken between 8am to 12 pm during the experiment. The increments were calculated from the differences of the two measurements (the first day and the last day of the experiment). Please see the revision.  

L139 WITH missing before spectrophotometry

Re: Added as suggested.

L151 support the choice of ANOVA with results of Shapiro-Wilk test, if the distribution diverses from the normal one, recalculate using Kruskal-Wallis test

Re: Good suggestion.  The Shapiro-Wilk test was used for the data analysis, and the original data were log-transformed to satisfy the normality and homoscedasticity assumptions of ANOVA. We have added it in the revision.   

L157 would be good to have correlation coefficient values here so that this colinearity would be obvious for the reader

Re: All soil TEB, CEC and BS were calculated by the four levels of N application effects on BCs (please see the methods) and the BCs was the important parameter of soil chemical properties.  Please see Figure 3 in the revision.

figures 2, 3 - bars represent mean, but SE is shown by WHISKERS - add this please

Re: it was corrected and indicated in figure legends in the revision.